# Persistence of Lockdown Consequences on Children: A Cross-sectional Comparative Study

**DOI:** 10.3390/children9121927

**Published:** 2022-12-08

**Authors:** Marina Picca, Paola Manzoni, Antonio Corsello, Paolo Ferri, Chiara Bove, Piera Braga, Danila Mariani, Roberto Marinello, Angela Mezzopane, Silvia Senaldi, Marina Macchi, Marco Cugliari, Carlo Agostoni, Gregorio Paolo Milani

**Affiliations:** 1SICuPP—Lombardia: Italian Primary Care Paediatrics Society—Lombardy, 20126 Milan, Italy; 2Department of Clinical Sciences and Community Health, University of Milan, 20122 Milan, Italy; 3Department of Human Sciences, State University Milano Bicocca, 20126 Milan, Italy; 4Fondazione IRCCS Ca’ Granda–Ospedale Maggiore Policlinico, Pediatric Area, 20122 Milan, Italy

**Keywords:** SARS-CoV-2 children, COVID-19 pediatrics, lockdown impact, distance learning, sleep disturbances toddlers

## Abstract

Lockdown during the COVID-19 pandemic had a significant psychological impact on children and adolescents. This study compared lockdown effects on children aged 1–10 years in 2020 and 2021. Two structured questionnaires were administered to 3392 parents in 2020, and 3203 in 2021. Outcomes considered for the data analysis included sleep changes, episodes of irritability, attention disturbances, distance learning and number of siblings. For data analysis, children were divided into two groups: pre-scholar (1–5 years old) and older ones. The lockdown was associated with a significant increase in sleep disturbances in 2020 and persisted after a year. The high prevalence of mood changes persisted unchanged in children under the age of 10 in 2020 and in 2021. Even if strengthened family ties seemed to mitigate the negative impact of lockdowns in 2020, this effect appeared absent or at least reduced in 2021. Irritability and rage in children were perceived to have increased in 2021 compared to 2020. A significant reduction in digital device use was observed in 2021 compared to 2020. Overall, the most harmful consequences of the lockdown in 2020 were still observed in 2021. Further studies are needed to analyze possible psychological effects that the generation who experienced the pandemic during early childhood may have, particularly in their future adolescence, in order to identify possible intervention practices to support families.

## 1. Introduction

On 23 March 2020, the COVID-19 disease, caused by the severe acute respiratory syndrome coronavirus 2 (SARS-CoV-2 virus), was declared a pandemic by the World Health Organization [1]. Evidence indicates that COVID-19 has a lower incidence and morbidity in the pediatric population compared to adults and elderly people, whereas the role of children in spreading the virus is still controversial [2]. To fight the pandemic, governments implemented disease containment measures. In Italy, one of the first interventions was school closure (21 February 2020), followed by social distancing and home quarantine. Schooling shifted to distance-learning models, and students were isolated at home for the rest of the academic year [3].

Although these measures might have a role in the fight against COVID-19, they also have negative consequences on children, affecting their social life, education, and mental health [4]. This disruption of several fields in children’s lives may have a long-term impact in the future.

For example, it has been difficult for some children to adapt to the new online learning system, which could lead to gaps in their school preparation and, therefore, in their performance in subsequent years, translating into fewer opportunities [5]. Moreover, isolation at home and quarantine have led to a change in the daily habits of the children, with an increase in screen time, consumption of junk food and sugary drinks and a reduction in physical activity, which could lead to an increased risk of cardiovascular disease over the years [6]. Finally, the psychological effects of the SARS-CoV-2 epidemic on the pediatric population are well known [7,8]. These include, among others, anxiety or depressive symptoms and posttraumatic stress disorder [9,10]. These effects might be linked to a change in sociability caused by the spread of the virus, with loss of peer interactions, social isolation, uncertainty, disruptions in daily routines and health concerns [11,12]. Moreover, the first lockdowns that suddenly occurred worldwide during the first months of 2020 played a significant role in this loss of social relations. These disorders could lead to an increased risk of depression and other psychiatric disorders in the future [13]. These changes may have a more significant impact in early childhood, when children start to explore the world outside their families, interacting with peers and attending their first school classes [14,15].

In a previous study conducted by our group, we investigated the alterations in the daily behavior of children aged between 12 months and 10 years that occurred after the first lockdown in Italy, between March and June 2020 [16]. In particular, we focused on those related to the possible changes in sleep quality, use of digital devices, mood and attention worsening. However, it is still unknown if such changes observed among children during the first lockdown tend to persist or modify [17]. Therefore, the purpose of this study is to evaluate whether such changes were still present among children one year after the lockdown in 2020. 

## 2. Materials and Methods

Two different cross-sectional questionnaires were administered during 2020 and 2021 in Lombardy (Italy), the most affected region in the first months of the pandemic, with 12 months in between. Surveys were elaborated by the Italian Society of Primary Care in Pediatrics (SICuPP) Lombardia, a group of researchers of the University of Milan Bicocca and the spin-off “Bambini Bicocca”. They were administered through an anonymous online platform distributed by SICuPP pediatricians to parents of children aged 1–10 years. Parents could fill in single or multiple-choice answers. The questions regarded children’s attitudes and behavior at home, sleep, familiar relationships, use of digital devices (time and purpose of usage), school activity and distance learning practices, and mood changes (rage, tantrums and irritability). We also collected information on the age and the gender of the child and the number of siblings in the family. Outcomes considered for the data analysis included sleep changes (“reduced/awakenings”, “improved”, “no significant change”), frequency of episodes of irritability, tantrums or rage, and attention disturbances. For data analysis, children were divided into two groups: pre-scholar (Group PS, between 1 and 5 years old) and older ones (Group OO, age 6–10 years). Data are presented as frequencies and percentages. The Fisher exact test and chi-squared test were used to compare data obtained from the first and the second questionnaire administration. The study was conducted in accordance with the principles of Helsinki’s declaration.

## 3. Results

Data from 3392 children were collected in 2020 (1688 in the group PS, 1704 in the group OO), while 3203 children were included in the 2021 survey (1401 in the group PS, 1802 in the group OO). No significant differences were present between the two years or the four groups in terms of age, gender, number of siblings or parental characteristics. Table 1 summarizes the most important outcomes observed, comparing data from the two questionnaires administered to the group PS and OO.

In the group PS, no significant differences emerged in sleep quality (73% and 72% in 2020 and 2021, respectively). On the contrary, while in the group OO the first lockdown of 2020 had a significant impact on sleep disturbances (34%), one year later this increase was reduced to 27% (*p* < 0.0001). Moreover, an increase in terms of attention disturbances have been reported only in the group OO (52% of children presented attention changes after the first lockdown, 59% one year later, with *p* < 0.0001).

Analyzing irritability/rage for the group OO and irritability/tantrums for the group PS, we observed persistence and an increase in these parameters during 2021. In particular, 51% of parents described a mood worsening in their children aged 6–10 years in the summer of 2020, which significantly increased (58%, *p* < 0.0001) in 2021. The same trend can be observed considering the irritability in preschool children (56% in the year 2020, 63% in 2021, *p* < 0.0001).

When considering the average time spent on digital devices during the day in younger children, no significant differences are observed comparing the first lockdown data with the average time one year later. In children attending schools, instead, significant and positive changes can be described if considering the purpose of use: when digital devices are used as entertainment for more than two hours per day, one year later a significant decrease can be observed (31% Vs. 14%, *p* < 0.0001), while the percentage of children using them for distance-learning and educational purposes for the same amount of time more than doubled (34% Vs. 80%, *p* < 0.0001). Furthermore, we observed how parents’ higher educational qualifications were associated with a shorter time spent using digital devices by their children. It is plausible to hypothesize that parents with higher educational qualifications, and therefore generally with a higher income, can offer their children a greater quantity of “external” supplementary activities (sports, music, workshops), and therefore their children spend less time on their own. Multiple logistic regressions for other parameters did not show significant trends or changes between the two years, and did not differ between the two groups.

## 4. Discussion

Even if the lockdown measures were effective in reducing the spread of the virus, the possible long-lasting impact it may have had on the physiological development and relations of children is not yet well known, nor if their social awareness may be altered in the long term [18,19]. This study shows for the first time that many negative changes observed among children during the lockdown in 2020 persist in the pediatric population after one year. However, many social habits, such as normal school and outdoor activities, have resumed.

In 2020, family dynamics profoundly changed, with a significant increase in remote working for parents and distance learning for children and adolescents [20,21]. It has not been described yet if and how another change of these family balances may have affected child behavior in the medium- or the long-term, nor if there could be differences when considering pre-school (1–5 years) or older children. Only a few studies about this topic are available, and they generally do not focus on the possible adverse effects that a prolonged lockdown may have on the emotional and socio-affective development typical of early childhood in the long term, nor on the possible persistence of uncertainty and worries a year later [22,23].

The lockdown also caused a significant increase in sleep disturbances, which seem to persist even after a year, especially in children under the age of 5. On the other hand, the high prevalence of mood changes persists unchanged in children under the age of 10, and parents have an increased perception of these disorders in 2021. The reduction of social activities, the discontinuity and the climate of uncertainty about the regularity of school attendance could have directly impacted their emotional state at home. A recent systematic review confirms our results, proving that children and adolescents experience more depressive and anxious symptoms than at the reported pre-pandemic rates [24].

Moreover, while younger children appear to be at lower risk of long-term effects on behavior and attention disorders, children who already had begun primary school tend to be more affected by emotional and behavior changes, the effects of which continue even after a year. The only positive effect that seems to persist in the long term could be linked to the use of digital devices for educational purposes. It is possible to hypothesize that using distance learning in a protected and supervised environment may represent a protective factor in preventing digital device misuse or abuse in daily life [25,26]. However, the percentage of children under the age of 5 who spend two hours or more a day with digital devices is not negligible (7% in 2020, 5% in 2021), considering that these subjects should use digital devices for no more than 1 h per day [27]. It should also be noted that digitalization of education can have negative effects. Particularly, it can lead to the phenomenon of technostress, the inability to cope with information and communication technologies (ICTs) in a healthy manner [28]. This phenomenon arises because of the feeling of being forced by ICTs to study/work faster and longer, and the overlap between home and workspaces, among others. The ability to handle technostress depends on individual coping resources and environmental factors [28]. Considering the heavy increase in the digitalization of education and work since the pandemic, more attention should be paid to finding ways to help children cope better with technostress [29].

The large sample represents the main strength of this study, which has been evaluated exactly one year after the first pandemic lockdown that involved the entire population in Italy, both adult and pediatric. Lombardy, in particular, represents the first and principal area hit by the SARS-CoV-2 epidemic, with an early and high disease burden [30]. Furthermore, our two samples are comparable both for socio-cultural and demographic characteristics. It should also be highlighted that although a year later many children were still in isolation and attended remote classes, many parents had returned to their work, thus causing a possible worsening of children’s social skills, especially in early childhood [23,31,32,33,34]. The main weakness of this study is that our data, although overlapping, do not correspond to the same sample, and still suffer from some bias. However, the high numerosity of the sample and the coincidence of many demographic characteristics reduce the risk of significant biases. One last weakness is represented by the fact that questionnaires did not consider the opinions of the children themselves, and were influenced by the perception of the single parent.

Our data confirm similar conclusions of other studies [6,13,35,36]. However, they represent a first assessment of a possible long-term impact of the pandemic on social skills in the pediatric age, maybe confirming that the protracted closure of schools had more negative than positive effects on children [37,38,39]. Moreover, even if in the first months of the pandemic the strengthened family ties had mitigated the negative impact of the children’s isolation [16,40], this effect seems to have decreased a year later.

## 5. Conclusions

Most effects observed following the first lockdown have not changed significantly after one year, persisting in children of both groups of age. Distance learning might have a positive effect, providing a higher use of personal devices for learning and interaction purposes without increasing their possible “solitary” use for leisure. Parents report a persistence of sleep disorders. Irritability and rage in children are perceived to have increased, probably due to the discontinuity and the climate of uncertainty about the regularity of school attendance and the loss of peer interactions. Further studies are needed to analyze possible psychological effects that the generation who experienced the pandemic during its early childhood may have, particularly in their future adolescence, in order to identify possible intervention practices to support families.

## Figures and Tables

**Table 1 children-09-01927-t001:** Answers about lockdown impact reported by parents, distributed by age groups.

			2020	2021
Group PS (≤5 years)	Sleep disturbances	Yes	460 (27%)	396 (28%)
		No	1228 (73%)	1005 (72%)
	Increased irritability/tantrums	Yes	940 (56%)	885 (63%)
		No	748 (44%)	516 (37%)
	Digital devices	2 h or less per day	1569 (93%)	1325 (95%)
		>2 h per day	119 (7%)	76 (5%)
Group OO (6–10 years)	Sleep disturbances	Yes	575 (34%)	484 (27%)
		No	1129 (66%)	1318 (73%)
	Lower attention levels	Yes	888 (52%)	1056 (59%)
		No	816 (48%)	746 (41%)
	Increased irritability/rage	Yes	870 (51%)	1038 (58%)
		No	834 (49%)	764 (42%)
	Digital devices (leisure)	2 h or less	1182 (69%)	1542 (86%)
		>2 h	522 (31%)	260 (14%)
	Digital devices (study)	2 h or less per day	1123 (66%)	360 (20%)
		>2 h per day	581 (34%)	1442 (80%)

## Data Availability

The datasets used and/or analyzed during the current study are available from the corresponding author on reasonable request.

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
