# Peer review of "Persistence of Lockdown Consequences on Children: A Cross-sectional Comparative Study"

_children, 2022, doi:10.3390/children9121927_

Round 1
Reviewer 1 Report
1) Please talk about the instruments used for data collection as well as statistical analysis in the abstract.
2) Keywords are much. Please use maximum of five keywords.
3) Please talk briefly about covid-19 disease in the beginning of the introduction.
4) Have you used any standard questionnaire for data collection?
5) Method section is relatively difficult to understand. Specifically, data analysis is not clear. How did you compare the changes during the time?
6) Are age or gender included into the regression analysis as covariate?
7) You have two conclusions. First, in the last paragraph. Then, in conclusion section separately. Why it is so?
Author Response
1) Please talk about the instruments used for data collection as well as statistical analysis in the abstract.
A: We agree with the reviewer, and we have better specified our methods in the abstract.
2) Keywords are much. Please use maximum of five keywords.
A: We thank the reviewer for this suggestion. We have reduced the number of keywords.
3) Please talk briefly about covid-19 disease in the beginning of the introduction.
A: We agree and thank the reviewer for his comment. We have extended the introduction, including a brief description of covid-19 widespread and its impact. We have included the most relevant references released on the subject.
4) Have you used any standard questionnaire for data collection?
A: The reviewer’s comment is pertinent. Despite already used and used for previous publication, the questionnaire employed for the study was not validated similarly to many others during the COVID-19 pandemic. We added such point to the limitation of the study.
5) Method section is relatively difficult to understand. Specifically, data analysis is not clear. How did you compare the changes during the time?
A: We appreciated the reviewer critique and clarified within the data analysis section of the methods the tests used for data comparison.
6) Are age or gender included into the regression analysis as covariate?
A: We thank the reviewer for this suggestion. We discussed this issue with the statistician. In the univariate analysis we have compared differences between the two groups about age and gender. No difference was found about these two factors. Therefore, we did not include such data to avoid an over-correction of the data.
7) You have two conclusions. First, in the last paragraph. Then, in conclusion section separately. Why it is so?
A: We thank the reviewer for his ideas and all the comments proposed. We better clarified our conclusions, widely modifying the discussion paragraph. Main changes are highlighted in red.
Reviewer 2 Report
This brief-report focus on a crucial topic.
It is a comparative study that analyses the persistence of the lockdown consequences on children in 2020 and 2021.
1. In my opinion, the introduction seems a little small. The information provided is "right to the point". It can be explored further and I believe it will benefit the article. Try to include more evidence on the impact that these issues may have on the future of children: For example, the disruption of the learning processes may impact future academic performance and translate into fewer opportunities. In fact, this pandemic may have influenced the future of many children in many ways. Try to explore that possibility according to available evidence on the topic.
That is explored in the discussion, however only regarding the lockdown topic in itself, which must be retained as it is. For this introduction part, there is evidence of the possible long-term impact that the disruption of any of the several fields in children's lives may have in the future, in an overall manner.
2. Table 1. Please remove the colours. There is no need for that distinction and it is not relevant to the rest of the study's data.
3. Discussion. " In conclusion, the only positive effect that seems to persist in the long term 136 could be linked to the use of digital devices for educational purposes. It is possible to 137 hypothesize that using distance learning in a protected and supervised environment may 138 represent a protective factor in preventing digital device misuse or abuse in daily life 139 [19,20]."
- After this sentence, I believe that a note of caution should be made regarding "technostress".
Author Response
This brief-report focus on a crucial topic.
It is a comparative study that analyses the persistence of the lockdown consequences on children in 2020 and 2021.
- In my opinion, the introduction seems a little small. The information provided is "right to the point". It can be explored further, and I believe it will benefit the article. Try to include more evidence on the impact that these issues may have on the future of children: For example, the disruption of the learning processes may impact future academic performance and translate into fewer opportunities. In fact, this pandemic may have influenced the future of many children in many ways. Try to explore that possibility according to available evidence on the topic. That is explored in the discussion, however only regarding the lockdown topic in itself, which must be retained as it is. For this introduction part, there is evidence of the possible long-term impact that the disruption of any of the several fields in children's lives may have in the future, in an overall manner.
A: We agree and sincerely thank the reviewer for his comment. We have extended the introduction, including a brief description of covid-19 widespread and its impact. We have included the most relevant references released on the subject. New introduction paragraph is now including the points stated by the reviewer. Main changes are written in red.
- Table 1. Please remove the colours. There is no need for that distinction and it is not relevant to the rest of the study's data.
A: The reviewer’s comment is pertinent. We have modified the table as suggested.
- Discussion. " In conclusion, the only positive effect that seems to persist in the long term 136 could be linked to the use of digital devices for educational purposes. It is possible to 137 hypothesize that using distance learning in a protected and supervised environment may 138 represent a protective factor in preventing digital device misuse or abuse in daily life 139 [19,20]."
After this sentence, I believe that a note of caution should be made regarding "technostress".
A: We appreciated the reviewer critique and clarified within the discussion, including a comments on technostress, its definition and some literature. Moreover, we better clarified our conclusions, widely modifying the discussion paragraph. Main changes are highlighted in red.
Round 2
Reviewer 1 Report
Thank you for your revision.